# Measurement of transverse emittance and coherence of double-gate field emitter array cathodes

Soichiro Tsujino[1], Prat Das Kanungo[1], Mahta Monshipouri[1], Chiwon Lee[1,2] & R.J. Dwayne Miller[2,3,4]

Achieving small transverse beam emittance is important for high brightness cathodes for free electron lasers and electron diffraction and imaging experiments. Double-gate field emitter arrays with on-chip focussing electrode, operating with electrical switching or near infrared laser excitation, have been studied as cathodes that are competitive with photocathodes excited by ultraviolet lasers, but the experimental demonstration of the low emittance has been elusive. Here we demonstrate this for a field emitter array with an optimized double-gate structure by directly measuring the beam characteristics. Further we show the successful application of the double-gate field emitter array to observe the low-energy electron beam diffraction from suspended graphene in minimal setup. The observed low emittance and long coherence length are in good agreement with theory. These results demonstrate that our all-metal double-gate field emitters are highly promising for applications that demand extremely low-electron bunch-phase space volume and large transverse coherence.

[1] Laboratory for Micro- and Nanotechnology, Department of Synchrotron Radiation and Nanotechnology, Paul Scherrer Institut, Villigen CH-5232, Switzerland. [2] Max Planck Institute for the Structure and Dynamics of Matter, Luruper Chaussee 149, Hamburg 22761, Germany. [3] Department of Chemistry, University of Toronto, 80 St. George Street, Toronto, Ontorio, Canada M5S3H6. [4] Department of Physics, University of Toronto, 80 St. George Street, Toronto, Ontorio, Canada M5S3H6. Correspondence and requests for materials should be addressed to S.T. (email: soichiro.tsujino@psi.ch).

The micro- and nano-fabricated field emitter array (FEA)[1], combining the field emission from thousands to millions of nanotips, is a promising source for applications that demands high current such as for cathodes of vacuum electronic amplifiers[2], X-ray sources[3] or mass spectroscopy[4]. These sources are distinct from thermionic emitters requiring heating current or photocathodes demanding ultraviolet-laser excitation in that they are efficient and enable fast switching of electron emission using single-gate FEAs by applying electrical potential to an integrated electron extraction gate electrode ($G_{ex}$) on the order of 100 V. Short electron bunch generation with the duration down to 200 ps was demonstrated solely by gate potential switching in an electron gun, with the acceleration electric field reaching tens of MV m$^{-1}$ (refs 5–7). Even faster switching of the electron emission, down to ps to fs, has been studied via the field emission of electrons excited by near-infrared ultrafast lasers[8–10]. However, single-gate FEAs are normally difficult to use for applications that require not only high current but also high beam brightness. This is because the transverse electron velocity spread and the intrinsic transverse emittance, which measures the phase space spread of an electron bunch, is typically an order of magnitude larger[5,11] than that of the state-of-the-art photocathodes[12,13]. The large angular spread of the electric field at the nanometer-scale radius of curvature at the emitter tip apex causes this problem[14] yet has not deterred the application of single-tip field emitters as evidenced by their successful use in the high-resolution electron microscopy[15] because of the extremely small emission area of single-tip emitter. This is not the case for array emitters because of the large array area. Nevertheless, the small energy spread of field emission beams strongly suggests that an ideal on-chip collimation lens for individual beamlets could be used to reduce the emittance substantially[16–18]. Therefore, such a double-gate FEA has been intensely studied in the past for pixel array displays, radiation-tolerant image sensors to reduce the cross talks between pixels[19–23] or as a cathode for accelerators that demands high current and high brightness at the same time[24]. Such a cathode that can generate electron bunches with a small phase space volume will also be beneficial for high frequency/THz vacuum electronic amplifiers with micron-scale gain structures[25] and the dielectric laser accelerators[26], as well as for electron imaging and diffraction analysis of nano-size biological specimens[27,28].

One of the challenges for the practical realization of a high current and low emittance double-gate FEAs has been to diminish the influence of the collimation potential $V_{col}$ on the emission current. This issue arises as the application of $V_{col}$, which is the opposite polarity to the electron extraction potential, decreases the electric field $F_{tip}$ at the tip apex and quenches the emission current, unless the collimation gate ($G_{col}$) structure is suitably engineered[22,29]. Our approach is the combination of increasing the distance between the $G_{col}$ edge and the emitter tip apex and partly shielding the emitter tip from $G_{col}$ by the edge of $G_{ex}$[30]. In this way, we were able to demonstrate the reduction of the beam divergence by approximately a factor of 10 with minimal decrease of the emission current[27,30,31]. However, in the reported beam imaging experiment in a diode configuration, it was difficult to separate the propagation of the electrons from their acceleration. Therefore, an experiment that directly characterizes the evolution of the beam upon propagation is needed to evaluate the emittance with a sufficient precision. It was also not apparent if the double-gate structure could preserve the coherent nature of the individual field emission beamlets.

In literature, the high beam brightness and the small transverse emittance of needle-shape single-tip field emitters have been widely studied both theoretically and experimentally[32–35]. However, aiming at applications for high-resolution electron microscopy, previous works have mostly analysed the electron optical characteristics and the virtual source size of single-tip field emitters. Experimental and theoretical studies of the total energy distribution of field emission electrons[36–38] have also been well established. However, theoretical and experimental study of the transverse emittance and the average transverse beam energy has been rarely conducted[17,39].

Here we study the intrinsic transverse emittance and the average transverse beam energy of double-gate FEAs experimentally and compare the experiment with theory. We adopt two experimental methods. First, we characterize the beam parameters of the double-gate FEA beam with a direct current (DC) gun test setup[7,40] and evaluate the transverse emittance. Second, we apply our double-gate FEA for the measurement of low-energy electron diffraction (LEED) using a suspended graphene film as the sample and measure the transverse coherence length, which is used to evaluate the transverse emittance. We find that the on-chip gate electrode can reduce the intrinsic emittance of the FEA beam by a factor of 10, enabling the successful observation of the atomic diffraction from graphene. Finally, we calculate the theoretical intrinsic emittance by applying the standard theory of field emission for electrons that is in excellent agreement with experiment.

## Results

**Double-gate FEA**. Figure 1 shows the scanning electron microscopic micrographs of the double-gate FEA. Each Mo nanotip emitter had a pyramidal shape with approximately a 1.5-μm base size and a tip apex radius of curvature approximately equal to 5 nm (ref. 30). The emitters were supported on a metal substrate. Two FEAs with $10^4$ tip array cathodes aligned in a 1.13-mm-diameter circle with the emitter separations of 10 μm were fabricated on a same chip. Figure 2a shows the field emission characteristics between the current $I$ detected at a counter

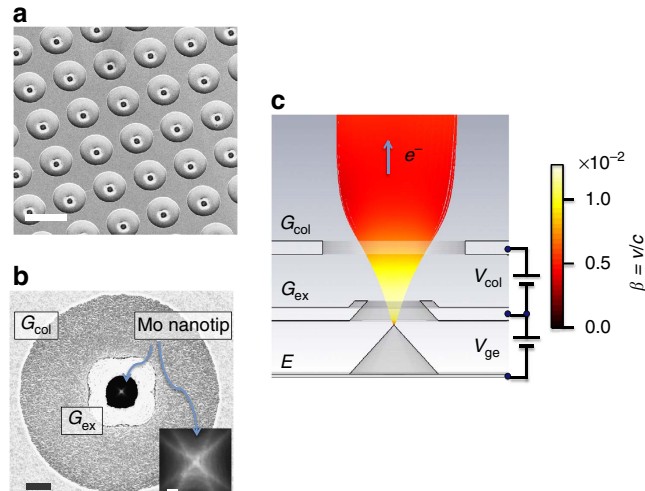

**Figure 1 | Double-gate FEA.** (**a**) Scanning electron microscopic (SEM) images of the fabricated double-gate Mo nanotip array cathode (bird's eye view). The scale bar is 10 μm. (**b**) Magnified SEM image of one emitter (top view). The scale bar (black) shows 1 μm. The inset is the top-view high-magnification image of one of the nanotip apex (the scale bar is equal to 20 nm). (**c**) Schematic cross-section of single emitter. The electron pulses are generated by the electron extraction potential $V_{ge}$ ($> 0$) and collimated by the collimation potential $V_{col}$ ($< 0$). $G_{ex}$: electron extraction gate, $G_{col}$: beam collimation gate, $E$: emitter substrate. The electron trajectories at the collimation condition simulated at the external acceleration field of 2.5 MV m$^{-1}$ are also shown. The colour code indicates the electron velocity $\beta$ ($= v/c$, where $c$ is the speed of light). The electron trajectories were calculated by a particle-tracking simulator (CST particle studio).

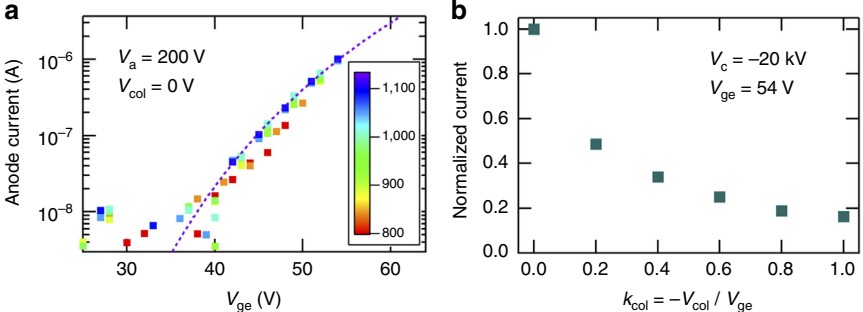

**Figure 2 | Field emission characteristics of double-gate FEA.** (**a**) The evolution of the field emission characteristic of the studied double-gate FEA device measured at low anode potential $V_a$ equal to 200 V, zero cathode potential and the zero collimation potential ($V_{col}$) displayed as the relation between the current $I$ collected at the anode and the electron extraction potential $V_{ge}$. The $I$–$V_{ge}$ scan was repeated by 1,311 times until the $I$–$V_{ge}$ characteristic became stable. Some of the scans between 801th and 1,311th of which scan numbers were denoted by colour bar are displayed. The dotted curve is the fitting. (**b**) The variation of the field emission current with the increase of beam collimation strength denoted by $k_{col} = -V_{col}/V_{ge}$ at $V_{ge}$ of 54 V, the cathode potential of $-20$ kV and the anode potential of 0 V.

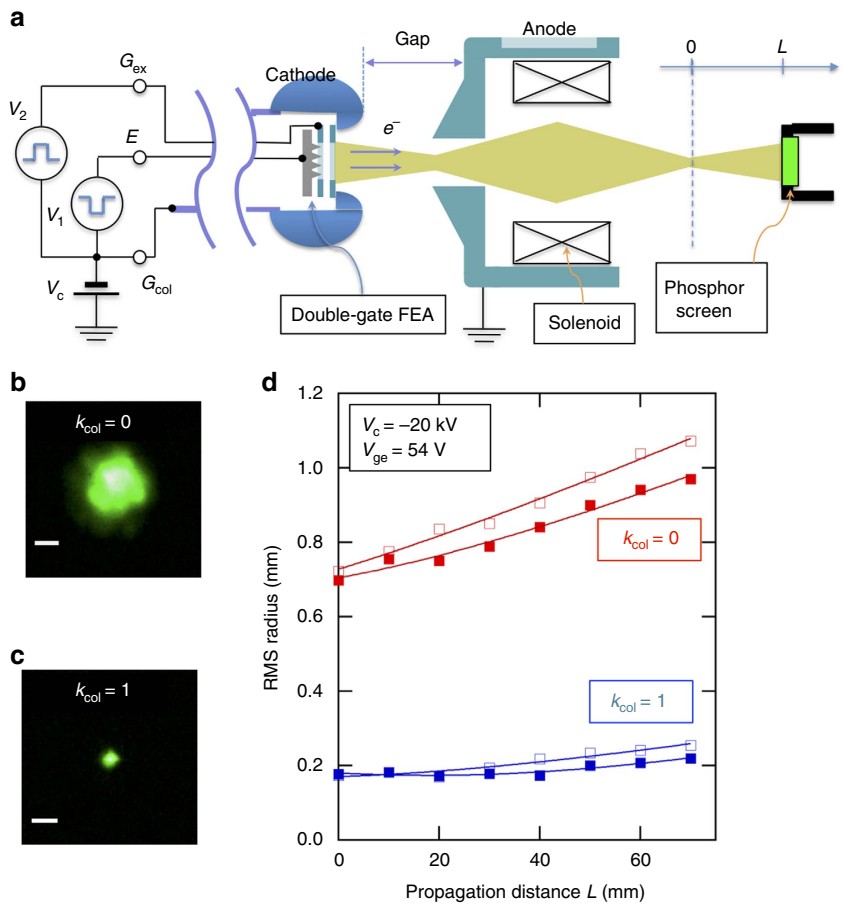

**Figure 3 | Measurement of intrinsic emittance.** (**a**) Schematic diagram of DC gun test setup. (**b**, **c**) Focussed beam image of the uncollimated ($k_{col} = 0$) and collimated ($k_{col} = 1$) beam at $V_{ge} = 54$ V and the cathode potential of $-20$ kV. The scale bars show 1 mm. (**d**) Variation of the r.m.s. radius with the propagation distance $L$ from the focal position ($L = 0$) for the uncollimated ($k_{col} = 0$, red) and the collimated ($k_{col} = 0$, blue) beams. The curves show the fitting results (see text). Filled and empty signs denote the radius in the horizontal and vertical directions, respectively. The solenoid current equal to 0.6 A and 0.65 A were applied for the beam focussing on the uncollimated beam and the collimated beam, respectively.

electrode (anode) and the electron extraction potential $V_{ge}$ applied between $G_{ex}$ and emitters $E$. This relationship was measured in the DC gun test setup[7,40], shown in Fig. 3a, with the nominal background pressure $<5 \times 10^{-9}$ mbar. The $I$–$V_{ge}$ characteristic is well described by the following equation[1,41–43],

$$I = A\left(V_{ge}/B\right)^n \exp\left(-B/V_{ge}\right); \qquad (1)$$

where $n$ is a parameter that is often close to but not necessarily equal to 2 when the image-charge lowering of the barrier is precisely considered[42,43]. When $n = 2$ is assumed as typically carried out in literature[1], we obtained a good fit with the experimentally measured $I$–$V_{ge}$ relation (dotted curve in Fig. 2a) with the parameters $A$ and $B$ equal to $0.82 \pm 0.24$ A and $495 \pm 12$ V, respectively. The exponential term in equation (1) is

unaffected by the precise treatment of the image charge effect[42,43] and gives the dominant contribution to the $I$–$V_{ge}$ relation. Therefore, by identifying it with the exponential term of the Fowler–Nordheim current density formula[41,42] given by $\exp(-b\phi^{3/2}/F_{tip})$, where $F_{tip}$ is the electric field at the emitter tip apex, $\phi$ is the molybdenum work function approximately equal to 4.5 eV, and $b = 6.830890\,eV^{-3/2}\,V\,nm^{-1}$ is a constant, $F_{tip}$ is obtained at a given $V_{ge}$. From Fig. 2a, we estimate $F_{tip} = 4.4 \pm 0.1\,GV\,m^{-1}$ at $V_{ge} = 54\,V$. When the collimation potential $|V_{col}|$, was increased, the emission current decreased, as shown in Fig. 2b in the case of $V_{ge} = 54\,V$. However, nearly 20% of the emission current observed for zero $V_{col}$ was retained at the maximal collimation condition of $k_{col} = 1$, as specified by the collimation parameter $k_{col} = -V_{col}/V_{ge}$ (ref. 30) (see Methods). The second double-gate FEA showed approximately the same emission characteristics.

**Emittance measurement in DC diode gun.** Next we applied a DC cathode potential of $-20\,kV$ and observed the image of the electron pulses emitted from the double-gate FEA by a phosphor screen when the beam was focussed by a solenoid at a position ($L = 0$) that was 100 mm downstream from the anode. To obtain the beam parameters, the r.m.s. beam size $\sqrt{x^2}$, the r.m.s. beam divergence $\sqrt{x'^2}$ and their correlation $\sqrt{xx'}$ at the position of $L = 0$, we moved the screen up to $L = 70\,mm$ and measured the evolution of the free propagation of the r.m.s. beam size $R_x(L)$. From that, we can evaluate the beam parameters by fitting with the formula,

$$R_x(L) = \sqrt{\langle x^2 \rangle + 2\langle xx' \rangle L + \langle x'^2 \rangle L^2}. \quad (2)$$

The intrinsic emittance is in turn evaluated from the beam parameters, the longitudinal velocity $\beta$ (divided by the light speed in vacuum $c$) and $\gamma = 1/\sqrt{(1-\beta^2)}$, as[44],

$$\varepsilon_x = \gamma\beta\sqrt{\langle x^2 \rangle \langle x'^2 \rangle - \langle xx' \rangle^2}. \quad (3)$$

As there is negligible correlation between beamlets emitted from different emitter tips[14], the intrinsic emittance of the double-gate FEA is written as

$$\varepsilon_x = \sigma_{s,x} \frac{\sqrt{\langle p_{s,x}^2 \rangle}}{mc}; \quad (4)$$

where $p_{s,x}$ is the momentum (in the $x$ direction) at the FEA, $\sigma_{s,x}$ is the r.m.s. FEA radius that takes into account the actual distribution of electron emission from the array emitters and $m$ is the electron rest mass. Because of the conservation of the intrinsic transverse emittance[44], we can determine the intrinsic emittance per unit FEA radius and the average transverse energy $\langle E_t \rangle = \langle p_x^2 + p_y^2 \rangle/(2m)$ from $\varepsilon_x$ and $\sigma_{s,x}$.

In Fig. 3b,c, we show the focussed double-gate FEA beams in uncollimated ($k_{col} = 0$) and collimated ($k_{col} = 1$) conditions. The fact that the beam at $k_{col} = 1$ was substantially smaller than the beam at $k_{col} = 0$ indicates the reduction of the emittance. This was indeed the case. As Fig. 3d shows, the uncollimated beam size increased from 0.7 to 1 mm after the free propagation of 70 mm. However, the collimated beam size was unchanged after the free propagation of the same distance. From the evolution of these beam sizes, we found that the emittance was equal to 1.1 μm at $k_{col} = 0$ and equal to 0.12 μm at $k_{col} = 1$ as summarized in Table 1. We found that the influence of the aberration of the solenoid lens on the evaluated emittance values is small (see Methods).

**Table 1 | Beam parameters of the double-gate metal nanotip array cathode at the beam waist at $L = 0$ for $k_{col} = 0$ (uncollimated) and $k_{col} = 1$ (collimated) beams with the cathode potential of $-20\,kV$.**

|  | $k_{col} = 0$ | $k_{col} = 1$ |
|---|---|---|
| $\sqrt{<x^2>}$ (mm) | $(7.04 \pm 0.14) \times 10^{-1}$ | $(1.80 \pm 0.05) \times 10^{-1}$ |
| $<xx'>$ (mm) | $(1.80 \pm 0.72) \times 10^{-3}$ | $(-0.12 \pm 0.07) \times 10^{-3}$ |
| $\sqrt{<x'^2>}$ | $(6.59 \pm 1.58) \times 10^{-3}$ | $(2.62 \pm 0.36) \times 10^{-3}$ |
| $\sqrt{<y^2>}$ (mm) | $(7.28 \pm 0.13) \times 10^{-1}$ | $(1.71 \pm 0.08) \times 10^{-1}$ |
| $<yy'>$ (mm) | $(3.03 \pm 0.68) \times 10^{-3}$ | $(0.08 \pm 0.10) \times 10^{-3}$ |
| $\sqrt{<y'^2>}$ | $(6.57 \pm 1.51) \times 10^{-3}$ | $(2.36 \pm 0.65) \times 10^{-3}$ |
| $\varepsilon_x$ (μm) | $1.21 \pm 0.28$ | $0.13 \pm 0.03$ |
| $\varepsilon_y$ (μm) | $1.05 \pm 0.28$ | $0.11 \pm 0.03$ |
| $\varepsilon$ (μm) | $1.13 \pm 0.28$ | $0.12 \pm 0.03$ |

$\varepsilon$ is the geometrical average of $\varepsilon_x$ and $\varepsilon_y$.

In Fig. 4, we show the cathode image at $L = 0$ for 40 keV beam energy, as detected by reducing the solenoid current by 10% from the focussing condition. The granular pattern, that is typical for FEAs without neon gas conditioning[7,45], indicated that not all emitters were active. The effect of the beam collimation was visible as the narrowing of the beamlet spots by increasing $k_{col}$ from 0 (Fig. 4a) to 1 (Fig. 4b). The spot narrowing was accompanied by the increase of the maximum intensity. In Fig. 4, we identified the circular envelope $S$ (chain curves in Fig. 4) as the outer boundary of the array and evaluated the ratio $f$ of the effective source size to the physical array size from the ratio of the integrations within the enclosed area as,

$$f_{s,x} = \frac{\int_S d\mathbf{r}(x-x_0)^2 P(\mathbf{r})/\int_S d\mathbf{r} P(\mathbf{r})}{\int_S d\mathbf{r}(x-x_0)^2/\int_S d\mathbf{r}}; \quad (5)$$

where $P(\mathbf{r})$ is the beam intensity distribution in Fig. 4 and $x_0$ is the centre of the beam determined from $P(\mathbf{r})$. We obtained $\sigma_{s,x}$ by multiplying $f_{s,x}$ to the lithographic r.m.s. array radius $\sigma_0$ of 0.28 mm. The results are summarized in Table 2. We found that the intrinsic emittance at $k_{col} = 1$ is $0.49 \pm 0.13$ μm (mm-r.m.s.)$^{-1}$. The corresponding average transverse beam energy $\langle E_t \rangle$ given by $\langle p_x^2 + p_y^2 \rangle/(2m)$ was equal to $0.12 \pm 0.06$ eV. In the case of uncollimated beam at $k_{col} = 0$, the emittance was $4.5 \pm 1.1$ μm (mm-r.m.s.)$^{-1}$ and $\langle E_t \rangle$ was equal to $10.3 \pm 5.1$ eV. The low emittance of the collimated beam is in good agreement with the values inferred from the previous beam imaging experiment of our double-gate FEAs[27,30,31], indicating the reproducibility of the results obtained in this measurement. As shown below, we obtained further support of the intrinsic emittance value from the electron diffraction experiment.

**Low-energy electron diffraction of graphene.** Next we measured the transmission of the FEA beam through a suspended single-layer of graphene. Figure 5b shows the result obtained from one of the samples when the collimation parameter $k_{col}$ of the incident beam was 1.0 and the beam energy was 1 keV. The direct beam, depicted in the right panel, was subtracted from the left image to highlight the hexagonal first- and second-order diffraction spots, centred at the point marked by the cross. This point coincided with the centre of the bright spot of the direct beam. The radial distance $R$ of the first-order diffraction spots from the centre equal to $6.04 \pm 0.19$ mm as well as that of the second-order diffraction spots agreed well with the value expected for the 1keV beam and the experimental conditions (see below and Supplementary Notes 1 and 2). The full width at the half

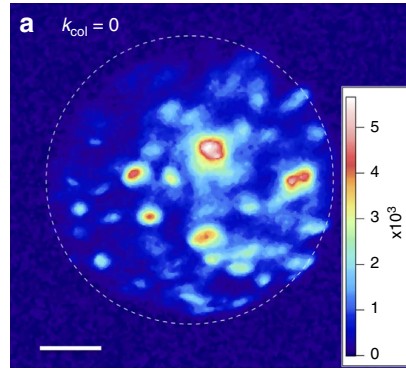

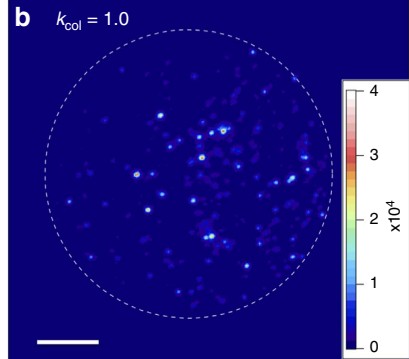

**Figure 4 | Double-gate FEA beam images. (a,b)** Beam image of the uncollimated ($k_{col} = 0$) and collimated ($k_{col} = 1$) condition, respectively, with $V_{ge}$ of 54 V observed at $L = 0$ and the cathode potential of $-40$ kV under the cathode imaging condition (solenoid current = 0.75 A). The scale bars indicate 1 mm on the phosphor screen. The field emission beamlets are finely visualized for the collimated beam ($k_{col} = 1$). The broken circle shows the envelope of the beamlet that corresponds to the physical boundary of the array, equal to 1.13 mm diameter on the chip.

**Table 2 | Summary of the r.m.s. beam size (fraction) at the source evaluated from the beam image, the r.m.s. transverse velocity, and the average transverse beam energy.**

|  | $k_{col} = 0$ | $k_{col} = 1$ |
|---|---|---|
| $f_{s,x}$ | 0.92 | 0.92 |
| $f_{s,y}$ | 0.86 | 0.83 |
| $(1/mc)\sqrt{p_{s,x}^2}$ | $(4.66 \pm 1.08) \times 10^{-3}$ | $(0.50 \pm 0.12) \times 10^{-3}$ |
| $(1/mc)\sqrt{p_{s,y}^2}$ | $(4.33 \pm 1.15) \times 10^{-3}$ | $(0.47 \pm 0.13) \times 10^{-3}$ |
| $\langle E_t \rangle$ | $10.3 \pm 5.1$ eV | $0.12 \pm 0.06$ eV |

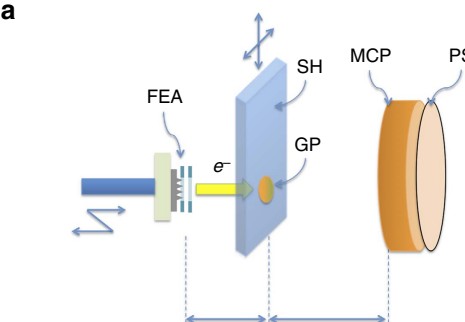

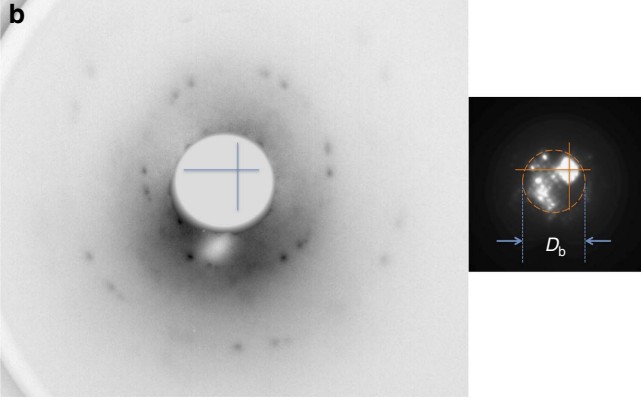

maximum $\Delta^{(1)}$ of the first-order diffraction spots on the screen were equal to $0.44 \pm 0.11$ mm and were smaller than the 1-mm-diameter bright spot of the direct beam on the screen (the right panel of Fig. 5b). Therefore we consider that $\Delta^{(1)}$ was determined by the transverse coherence length and the sample domain size that were much smaller than the direct beam and not by the beam spot size on the sample. Accordingly, we can estimate the transverse coherence length from the ratio $R/\Delta^{(1)}$ (ref. 46). To quantitatively evaluate the r.m.s. coherence length $\sigma_c$, we calculated this ratio as a function of $\sigma_c$ by a one-dimensional lattice model illuminated by Gaussian electron wave functions with $\sigma_c$ (see Supplementary Note 1, Supplementary Figs 1 and 2). Using this relation, we evaluated $\sigma_c$ by taking the value that corresponds to the experimental value of $R/\Delta^{(1)}$ equal to $13.7 \pm 3.5$ and found $\sigma_c = 0.89 \pm 0.25$ nm.

Given the intrinsic transverse emittance and the r.m.s. spot size of the beam on the sample, $\sigma_c$ is written as[47,48] (Supplementary Equation (3)),

$$\sigma_c = \frac{\hbar}{mc} \frac{\sigma_x}{\varepsilon_x}. \tag{6}$$

The r.m.s. spot size $\sigma_x$ on the sample was approximately equal to the cathode source size $\sigma_{s,x}$, as one can see in Fig. 5b that the beam envelope diameter $D_b$ corresponded approximately equal to 1 mm on the sample (determined from the shadow of the 80-μm-pitch, 300-mesh transmission electron microscopic (TEM) grid pattern). Therefore, we identify $\sigma_c$ given by equation (6) as the source coherence length of our double-gate FEA. Using the previously determined transverse emittance of the same double-gate FEA equal to $0.49 \pm 0.13$ μm (mm-r.m.s.)$^{-1}$ ($k_{col} = 1$ beam), we found $\sigma_c = (0.79 \pm 0.20)$ nm. This is in excellent agreement with the value evaluated from the diffraction spot size.

**Figure 5 | Low-energy electron diffraction of graphene using a double-gate FEA. (a)** Schematic of the low-energy electron diffraction experiment. The double-gate FEA (FEA) was separated from the graphene (GP) sample on a TEM grid with the distance of D1 = 7.1 mm (the distance 4.6 mm between FEA and the sample holder (SH) and the thickness of SH equal to 2.6 mm). The transverse position of GP was adjusted by a manipulator. The FEA beam transmitted through and diffracted by GP was detected by the phosphor screen (PS) after amplified by a single-stage multi-channel plate (MCP). GP and MCP were separated by D2 = 29 mm. GP and SH was biased at 1 kV. The GP side of MCP surface was biased at 250 V. **(b)** The electron diffraction of a graphene film generated with the maximally collimated beam ($k_{col} = 1.0$) produced from the double-gate FEA. The direct beam (right panel, same spatial scale with the diameter $D_b \approx 1$ mm on the sample) was subtracted and the intensity was factor of a 100 enhanced numerically. The centre of the hexagonal diffraction spots is denoted by a cross: a small graphene domain at the centre of the cross, which was smaller than the bright spot size, contributed the diffraction.

Similar to a previous report (for example, in ref. 49 Supplementary File), individual diffraction spots consisted of 2–3 separate peaks, indicating the sampling of several graphene domains with a few degrees of rotation. As shown in Supplementary Fig. 3, the radial distance of the diffraction spots from the centre was quantitatively correlated to the incident beam potential that determines the electron wavelength. Also observed was the rapid smearing of the diffraction spots with the decrease of $k_{col}$ from 1 as expected from the increasing $\langle E_t \rangle$ with the increase of $k_{col}$ (Supplementary Note 2 and Supplementary Fig. 4).

**Transverse energy of field emission beam.** We now compare the experimentally obtained $\varepsilon_x$ and $\langle E_t \rangle$ with theory. For this purpose, we use the fact that $\langle E_t \rangle$ is the difference of the average total electron energy $\langle E \rangle$ and the average normal energy $\langle E_z \rangle$ (where $E_z$ is the kinetic energy of the electron in the direction perpendicular to the emission surface) and refer to the result of Swanson et al.[37] for $\langle E \rangle$ and $\langle E_z \rangle$ calculated by the standard field emission theory in the case of metals with the Fermi energy $E_F$ and the work function $\phi$ much larger than the emitter temperature. We found that $\langle E_t \rangle$ is given by the exponential slope $d_F$ of the transmission function $T(E_z)$ at the Fermi energy $E_F$ that is mainly determined by the work function and the electric field at the tip ($F_{tip}$) as[42],

$$\frac{1}{d_F} \equiv \frac{\partial}{\partial E_z} G(E_z) \bigg|_{E_z = E_F} = \tau_F \frac{3b\phi^{1/2}}{2F_{tip}}; \qquad (7)$$

where $\tau_F$ is close to 1[42,43] and $b$ is a constant (given below equation (1)). As shown in ref. 42, the function $G(E_z)$ is derived from $T(E_z)$, which is written by a prefactor $P(E_z)$ and the exponential of $G(E_z)$ as,

$$T(E_z) = P(E_z) \exp[-G(E_z)]. \qquad (8)$$

In equation (8), $P(E_z)$ is a weakly varying function of energy, and its derivative is negligible in comparison to the derivative of the exponential term. Therefore $P(E_z)$ is safely neglected (see also Supplementary Note 3). By using the value $F_{tip} = 4.4 \pm 0.1\,\text{GV m}^{-1}$ evaluated from the fitting of $I$–$V_{ge}$ characteristic, we found $d_F = 0.19 \pm 0.01\,\text{eV}$. This is in good agreement with $\langle E_t \rangle$ obtained

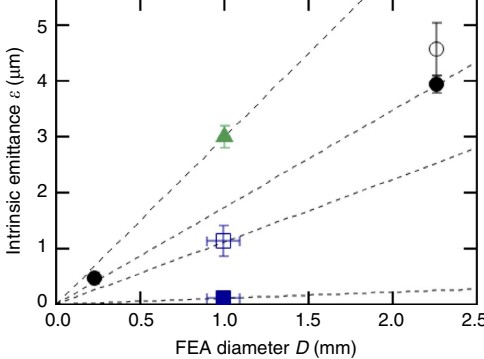

**Figure 6 | Summary of intrinsic FEA emittance.** The emittance $\varepsilon$ denotes normalized r.m.s.[44] values. The filled square ($k_{col} = 1$) and the empty square ($k_{col} = 0$) are the present double-gate FEA results. The intrinsic emittance values of single-gate FEAs are also shown. The circles show $\varepsilon$ of single-gate FEA with molded nanotips fabricated as the double-gate FEA in the present report. The data denoted by filled circles were measured by solenoid scan and the error bar indicates the r.m.s. fitting uncertainty, see ref. 5. The value denoted by empty circle was measured by inserting a single slit mask[40] and the error bar indicates the r.m.s. uncertainty of the evaluation of the beam divergence. The triangle is the result of a Spindt FEA reported in ref. 40 by solenoid scan. The lines are guide for the eye: representing $\varepsilon = \xi D$, with $\xi$ equal to 0.1, 1.1, 1.7 and 3.0.

from our experiments. The theory predicts higher emittance and higher $\langle E_t \rangle$ at increased emission current with larger $F_{tip}$. However, because of the exponential sensitivity of the emission current, the actual increase of the emittance and $\langle E_t \rangle$ are expected to be small: for example, at the emission current two orders of magnitude higher than the present experiment, the required $F_{tip}$ is about 30% higher than the value above, hence the expected increase of $\langle E_t \rangle$ and $\varepsilon_x$ are approximately 30% and 15%, respectively. We note that in ref. 39, Shimoyama and Maruse theoretically analysed the intrinsic axial brightness of field emitters, which approximately corresponds to the case when the geometrical distribution of $F_{tip}$ and the resulting transverse velocity spread were eliminated. Although the energy scale that determines the intrinsic axial brightness is different from that of the transverse emittance, their result is same as the quantity calculated as the difference of $\langle E \rangle$ and $\langle E_z \rangle$ because of the functional form of the approximated transmission function for metals.

**Discussion**
We demonstrated that, despite the large initial angular spread of FEA beam owing to the geometrical distribution of the tip apex electric field, the intrinsic transverse emittance could be reduced to $0.49\,\mu\text{m}\,(\text{mm-r.m.s.})^{-1}$. This is comparable to the thermal emittance of the state-of-the-art ultraviolet-laser-excited photocathode equal to $0.4$–$0.6\,\mu\text{m}\,(\text{mm-r.m.s.})^{-1}$ (refs 12,13). This was achieved by integrating a second electrode for focussing individual beamlet. We confirmed this experimentally by measuring the free propagation of the beam and via the measurement of the transverse coherence length in a LEED measurement of a freestanding graphene.

In Fig. 6, we summarize the observed intrinsic emittance (geometrically averaged for the horizontal and vertical directions) of the uncollimated ($k_{col} = 0$) and the collimated ($k_{col} = 1$) beam in the present work, together with the intrinsic emittance of single-gate Mo FEAs reported previously. The circles (filled and empty) (ref. 6) show the values of single-gate FEAs with emitters and the gate aperture fabricated by the same method as the emitters and $G_{ex}$ of the double-gate FEA. The triangle shows the emittance of a Spindt FEA[40]. Figure 6 shows that the emittance of the uncollimated beam of the double-gate FEA is the same as that of the single-gate FEAs of the same type within a factor of 1.5. Comparing to these previous results, the significance of the on-chip beam collimation on the reduction of emittance is apparent. The reduction of the emittance was sufficient to increase the transverse coherence length of the FEA beam and to enable the observation of the electron diffraction from a graphene film. Recently, Kirchner et al.[48] reported the source transverse coherence equal to $0.79\,\text{nm}$ for a ultraviolet-laser-excited photocathode developed for ultrafast electron diffraction. Their value is same as that of our double-gate FEA. In their experiment, they achieved the transverse coherence length of $20\,\text{nm}$ at the sample position by expanding the beam by 25 times[48]. Such technique is readily applicable with our FEA for future LEED experiments to study biological macromolecules.

For high-frequency vacuum electronic amplifier applications, the transverse beam energy on the order of $0.1\,\text{eV}$ implies that the required minimal magnetic field for beam transport through the micron-scale gain structure for THz frequency range is much smaller than $10\,\text{kG}$, which is achievable with permanent magnets[25]. Therefore, the double-gate FEAs may offer a possible solution to realize a high-power THz sources using such technologies. The double-gate FEA will also be beneficial as an electron source that requires large transverse coherences as the successful LEED experiment demonstrated in the present work,

as well as applications that require low-phase space spread of the electron bunches such as the electron injector for the dielectric laser accelerators to minimize current loss[26,50].

The observed low emittance and the low $\langle E_t \rangle$ of 0.12 eV shows the importance of low resistivity of the emitter tips and the substrates to produce an FEA beam with the lowest emittance/transverse energy by way of double-gate structure. This is because the finite resistivity would induce a non-uniform beam potential distribution when the electron bunches pass through the collimation gate aperture and would result in a distribution of $k_{col}$ among those emitters. To avoid premature failure of FEA at low emission current owing to the emission from the statistically sharpest emitters, such finite resistivity of the cathode material or substrate has been intentionally introduced for reported high current-emitting single-gate FEAs. The effectiveness of such ballast resisters requires voltage drops of 1–10 V or higher. However, such a distribution of the beam potential will limit $\langle E_t \rangle$ to the same amount. Consequently, to prepare a double-gate FEA for high current emission, other strategies such as the careful increase of emission current as shown by Schwoebel et al.[51] or the neon-gas conditioning[45], is better suited.

The standard theory of field emission from metals predicted an $\langle E_t \rangle$ that agreed with experiment fairly well. Interestingly, the theory predicted that $\langle E_t \rangle$ is unaffected by an increase in temperature when it is approximately $< 2{,}000$ K ($\approx d_F/k_B$,), even though the increase of $E$ and $E_z$ with the increase of temperature is substantial[38]. This suggests that low $\langle E_t \rangle$ may be maintained at increased transient electron temperatures, for example, under intense laser excitation for producing ultrafast electron pulses[8]. We also note that, by examining the theory, $\langle E_t \rangle$ can be much smaller than $d_F$ determined by equation (8) for materials with the Fermi energy much smaller than 10 eV but with the same work function: a cathode with the Fermi energy of 0.6 eV will have a factor of 4 lower $\langle E_t \rangle$ than a cathode with the Fermi energy of 5–10 eV at the same $F_{tip}$ and $\phi$. Even lower $\langle E_t \rangle$ is predicted by this theory for cathodes with lower $E_F$. However, the reduction of $E_F$ also accompanies the reduction of the field screening length, hence the enhanced band bending by the tip electric field, the reduction of the current density and enhanced temperature dependence of $\langle E_t \rangle$). Parametric study of these different effects that takes into account these trade-offs will be needed to find a material with optimal performance.

## Methods

**Double-gate FEA.** The cathodes were fabricated by a molding method for the production of the molybdenum emitters, the self-aligned method for $G_{ex}$ fabrication and the electron-beam lithography method for $G_{col}$ fabrication[30]. We used a $10^4$-tip array cathode aligned in a 1.13-mm-diameter circle with 10-μm separations. Each emitter was a 1.5-μm base pyramidal shape with the apex radius of curvature in the order of 5 nm. The emitter shape was determined by the anisotropic etching of the Si mold wafer and its subsequent multiple oxidation prior to the deposition. The nanoscale tip shape was engineered by repeated oxidation of the mold wafer prior to the sputter deposition of the cathode material (Mo). $G_{col}$ and $G_{ex}$ consist of 300-nm-thick molybdenum films. The emitter substrate, $E$, and $G_{ex}$ were separated by a 1.2-μm-thick $SiO_2$, and $G_{col}$ and $G_{ex}$ were separated by a low-stress 1.2-μm-thick SiON. The nominal diameters of the gate apertures were 1.2 μm for $G_{ex}$ and 7 μm for $G_{col}$, respectively. The molybdenum emitter array was supported on a 300-μm-thick electro-plated nickel substrate with the resistivity well below 0.1 mΩ cm. Therefore, different from Si FEAs or Spindt-type FEAs fabricated on silicon substrates, the resistive voltage drop between emitters or in the substrate at finite emission current is negligible and the RC constant of the tip is in picosecond range (limited by the resistance of the gate layers) allowing for the sub-nanosecond direct switching by gate pulses independently from the acceleration[5–7].

**DC gun test setup.** We used the DC diode gun test setup[7,40] to perform the transverse beam parameter measurements of the double-gate FEA. The setup was used previously to measure the transverse emittance and to test the 200-ps electrical switching of single-gate FEAs[7,40]. To load single-gate FEAs in the gun[7], the chip was sandwiched by the cathode cap, which faced the anode, and a spring

contact connected to the centre conductor of the coaxial feed-through. The electron pulses were produced by applying negative potential pulses with the amplitude of $-V_{ge}$ to the emitter substrate $E$ with respect to the cathode cap. The electron pulses were extracted through the 4-mm-diameter iris of the cathode cap. The vacuum flange that was electrically connected to the cathode cap and holding the FEA was electrically insulated from the anode side of the gun. Therefore, by applying a negative DC high voltage to this flange, the FEA pulses were accelerated. The gap between the cathode and the anode, that was chosen to be 8 mm in the experiment, can be varied in situ between 4 and 15 mm. The electron pulses went through the anode iris with the diameter of 1.5 mm. The electron pulses were then refocussed by the solenoid integrated in the anode block.

To integrate double-gate FEAs, we modified the single-gate FEA holder to allow for an additional electrical contact. $G_{col}$ was in contact with the cathode cap, and the emitter substrate $E$ and $G_{ex}$ were connected, respectively, to both the inner and the outer conductor of a spring-loaded coaxial contact pin, which was in turn connected to an insulated coaxial electrical feed-through. For the connection to $G_{ex}$, FEA was placed on a custom-made ceramic chip carrier with a Au-plated patterned contact, to which $G_{ex}$ was wire-bonded. To produce electron pulses, we applied two synchronized and balanced voltage pulses, $V_{ge}$ ($>0$) between $G_{ex}$ and $E$ and $V_{col}$ ($<0$) between $G_{col}$ and $G_{ex}$ by using a custom-built double voltage pulser with the rise and fall time of 100 ns.

After loading the FEA into the setup and evacuating the chamber until the base pressure below $5 \times 10^{-9}$ mbar was reached, the FEA was conditioned by repeatedly measuring the field emission ($I$–$V_{ge}$, where $I$ is the current measured on the anode and $V_{ge}$ is the electron extraction potential) characteristic until it became stable as shown in Fig. 3a. For this purpose, we applied 200 V to the anode block and 0 V to the cathode flange and cycled the electron extraction potential $V_{ge}$ between 0 V and a certain value that was slowly increased until 54 V in time. At this low acceleration voltage, the anode block captured all the electrons. They did not go through the iris. After approximately 1,000 scans, the $I$–$V_{ge}$ characteristic became stable and well represented by the Fowler–Nordheim equation as shown in Fig. 3a.

For the measurement of the beam image and collimation characteristics, we applied the cathode potential of $-20$ kV, connected the anode to ground potential and applied pulsed gate potential with $V_{ge} = 54$ V and $V_{col} = -k_{col} V_{ge}$ with $k_{col}$ between 0 and 1. The electron pulses produced from the FEA were slightly focussed and went through the 1.5-mm-diameter anode iris without loss at this cathode potential. Subsequently, the electron pulses with the fixed beam energy were refocussed by a solenoid integrated in the anode block and freely propagated to a phosphor screen. The zero position of the phosphor screen ($L = 0$) was 100 mm downstream from the exit of the anode block. From $L = 0$, the phosphor screen position was moved by a linear translation stage away from the anode. A synchronously triggered charge-coupled device camera recorded the beam image detected on the phosphor screen. In the experiment with the cathode potential of $-20$ kV, the acceleration electric field $F_{acc}$ at the FEA surface was equal to 1.3 MV m$^{-1}$. $F_{acc}$ increases to 2.5 MV m$^{-1}$ midway between the cathode and anode. The pulse duration was equal to 1.5 μs. To increase the image signal-to-noise ratio, we applied multiple pulses ($<200$ shots) with the period of 50 μs. From thus observed beam images and their evolution with the variation of $L$ by free propagation, the intrinsic emittance was evaluated as described in the main text.

We note that due to the aberration of the solenoid lens, thus obtained intrinsic emittance is the upper limit of the actual value. However, the observed small field emission beamlets (Fig. 4b) of the maximally collimated beam in beam imaging mode suggests that the nonlinearity of the solenoid lens is small. We also estimated the aberration in the actual measurement condition and found that it is in fact small: The solenoid current of 0.57–0.6 A and the peak solenoid field of 0.03 mT was orders of magnitude smaller than the saturation field of the iron core, therefore the nonlinearity of the solenoid focussing due to the saturation of the iron core is neglected. At the beam energy of 20 keV and the average transverse energy of the maximally collimated beam energy of 0.1–0.2 eV, the chromatic aberration of the solenoid can be neglected ($<10^{-4}$). To estimate the first-order spherical aberration of the solenoid, we apply the formula[52],

$$\Delta\varepsilon_{sp} = \gamma\beta \frac{C_1}{f} \left[ \langle r^2 \rangle \langle r^6 \rangle - \langle r^4 \rangle^2 \right]^{1/2}; \tag{9}$$

$$C_1 = \frac{1}{2} \frac{\int \{dB/dz\}^2 dz}{\int B^2 dz}; \tag{10}$$

$$\frac{1}{f} = \left( \frac{e}{2m\gamma c\beta} \right)^2 \int B^2 dz; \tag{11}$$

where $B(z)$ is the magnetic field along the beam axis ($z$ is the beam propagation direction) reported in ref. 53. We multiplied $\gamma\beta$ in equation (9) to compare it with the normalized r.m.s. emittance $\varepsilon_{k=1} = 0.13$ μm of the maximally collimated FEA beam reported in the main text. Assuming a Gaussian beam profile with the r.m.s. beam radius of 0.5 mm in the solenoid, we found $\Delta\varepsilon_{sp} = 0.027$ μm. This is smaller than $\varepsilon_{k=1}$ by more than a factor of 4.

**Low-energy electron-diffraction chamber.** The LEED experiment was conducted in a setup depicted in Fig. 5. We used the second double-gate FEA fabricated on a

same chip and with approximately same field emission characteristics as the first double-gate FEA used in the DC gun experiment described above. We used monolayer graphene samples, sample 1 and sample 2, suspended on TEM grids: sample 1 on a holy amorphous carbon on the grid and sample 2 on an amorphous carbon film with 2-μm holes aligned with 4-μm pitch. Samples 1 and 2 were purchased from Graphenea (CVD graphene transferred on a TEM grid, Au-QUANTIFOIL R 2/4) and TED TEPLA Inc. (PELCO Graphene TEM Support Films), respectively. We observed Bragg reflections from both samples. Figure 5 shows the result measured with the sample 1 and the results from sample 2 is shown in Supplementary Note 2. The results indicated the observation of Bragg reflections from multiple domain graphene lattices with stronger first-order spots. The graphene-on-TEM grid samples were held on a 2.6-mm-thick Aluminum plate over a 2-mm-diameter hole. The electron beam was irradiated on the graphene through the hole. The FEA beam was accelerated to 1 keV when it irradiated the graphene by applying the same potential to the graphene sample holder. The transmitted and reflected electron beam propagated toward the electron detector. The FEA and the Aluminum plate were separated by 4.5 mm, hence the acceleration field of the field emission beam was $0.22\,MV\,m^{-1}$. The electron detector consisted of a single-stage multi-channel plate biased at 500 V for amplification and a phosphor screen biased at 4.5 kV. The entrance surface of the detector was biased at 250 V. We recorded the beam image detected on the phosphor screen by a synchronously trigged charge-coupled device camera. We produced the FEA beam by applying $V_{ge}$ of 47 V and $k_{col}$ of close to and equal to 1. Higher $k_{col}$ resulted in the substantial decrease of emission current and loss of the enhancement of the beam intensity. Separate measurement with smaller acceleration of $0.1\,MV\,m^{-1}$ without the graphene sample suggested that the beam spot of $k_{col} = 1$ beam on the TEM grid was approximately the same size as the FEA array (diameter of 1.13 mm) as assumed in the main text. This was compatible with the estimated FEA beam size on the TEM grid displayed in Fig. 5. The graphene sample and the front plane of the electron detector were separated by 27 mm. We applied 0.9 ms gate pulses to the FEA to produce the collimated FEA beam. Single-shot images were sufficient to resolve the diffraction spots by $k_{col} = 1$ beam; nevertheless, the data displayed in Fig. 5 was averaged over 200 pulses to improve the signal-to-noise ratio. By moving the FEA beam in the transverse direction, we aligned the graphene position with respect to the FEA beam where the diffraction spots were the brightest. As shown in Fig. 5, this was when the centre of the diffraction spots was shifted from the apparent centre of the FEA beam. As described in Supplementary Note 2, we observed that when the potential on the incident surface of the electron detector was zero, the distance from the diffraction spots to their apparent centre was equal to the value calculated from the beam potential ( = the potential applied to the sample holder) and the distance between the graphene and the electron detector. Hence, together with the hexagonal symmetry of the observed spots, we concluded these arise from electron diffraction from the graphene.

**Data availability.** The data that support the findings of this study are available from the corresponding author upon request.

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

## Acknowledgements

We thank M. Paraliev for technical support for preparing a double-gate FEA holder for DC gun and the pulser; C. Zumbach, D. Hauenstein and S. Stutz for their assistance to construct the FEA holder; V. Guzenko, J. Lehmann and K. Vogelsang for their assistance for FEA fabrication; M. Aiba for critical reading of the manuscript and helpful comments; A. Streun for helpful discussions on the nonlinearity and aberration of solenoid lens; Y. Gotoh for discussions on the field emission electron energy; J. Faist for helpful discussion on the tunneling; D. Dowell for helpful discussion on the emittance theory; G. Kassier for helpful discussion on the relation between emittance and coherence; H. Yamada for his support on graphene preparation; R. Erni and E. Müller for helpful discussion on the electron diffraction of graphene; and J. Gobrecht and H.-H. Braun for their support. This work was partially supported by SwissFEL project at the Paul Scherrer Institute and Swiss National Science Foundation, Grant Nos. 200020_143428 and 2000021_147101, and the Max Planck Society.

## Author contributions

S.T. and R.J.D.M. conceived the experiments. S.T. and P.D.K. fabricated the FEAs. S.T. conducted and analysed the emittance experiment. S.T., M.M. and C.L. conducted the LEED experiment. S.T. wrote the manuscript. All the authors read the final manuscript.

## Additional information

**Competing financial interests:** The authors declare no competing financial interests.

