## [Peer Review File · Nature Communications]

Reviewers' comments:

Reviewer #1 (Remarks to the Author):

My recommendation is not to publish the paper in Nature Communications because the same work has already been reported by the authors, at least in part, in a recent conference that has a technical digest copyrighted by IEEE and that will be widely available very soon through IEEE Xplore.

At the 29th International Vacuum Nanoelectronics Conference (IVNC 2016), which took place in Vancouver Canada, July 11-15, the same authors of the Nature paper presented the oral paper FE5.C2 "Intrinsic Emission and Coherence of Double-Gate Field Emitter Arrays". The Technical digest of the conference has on pages 77 and 78 the extended abstract of such oral paper. The technical digest of IVNC 2016 was made available to all attendees to the conference (~120 people as I understand) and very soon will be made available to anyone through IEEE Xplore. To the best of my understanding publishing beforehand, fully or in part, work in a widely disseminated source is against the policies of Nature Communications.

The website of the conference IVNC 2016 clearly states that the extended abstracts will be available in IEEE Xplore and that are copyrighted material of IEEE (<http://www.vacuumnanoelectronics.org/extended-abstract-submission/>). Quoting from such website:

"Please note that completing and submitting the copyright form, as well as ensuring that the format of your extended abstract complies with IEEE requirements, are mandatory in order for your extended abstract to be published in IEEE Xplore. Therefore, please follow the instructions given on this page carefully, and strictly adhere to all aspects of the provided template. Extended abstracts not complying with IEEE requirements or those with missing or incomplete copyright forms, will be rejected by IEEE and not appear in the conference proceedings on IEEE Xplore."

The matter is a lot more serious. You used in your Nature paper arguably the same figures you used in your IVNC paper, which to the best of my knowledge infringes copyrights of IEEE. Specifically:

(i) The figure 1, left in the IVNC paper is fundamentally Figure 1 (a) and (b) of the Nature paper. Yes, there is a different pictorial composition, there is a new scale bar, the close-up of the tip is included, but those are minor differences. The fact is that the SEMs show the same devices, in exactly the same orientation, and even the same defects in the array structure. The data are the same.

(ii) The figure 1, center in the IVNC paper is identical to Figure 3 (b) and (c) in the Nature paper. Again, there are new scale bars but the actual images are pixel by pixel the same. The data are the same.

(iii) The figure 1, right in the IVNC paper is the negative of Figure 5 (b) in the Nature paper. Again, the change from positive to negative is unimportant; the information, pixel by pixel, is the same. To the best of my understanding the two figures show the same experiment. The data are the same.

In conclusion, all figures in the IVNC paper were reused in the Nature paper, although they have IEEE copyright, they were already published, they show the same data, and none of this is acknowledged in the Nature paper.

To the best of my knowledge you did not disclose to Nature that you wrote such conference paper, or that there is an existing IEEE-copyrighted proceedings document about to become widely

available through IEEE Xplore, and you did not reference the IVNC paper in your Nature paper. All these facts make me wonder if you were trying to hide the existence of the IVNC paper from the reviewers.

Since the work is at least in part already published and copyrighted, the publication in Nature Communications should be denied. This precedes any pondering on the merits of the work, which at this point is a worthless exercise.

Reviewer #2 (Remarks to the Author):

The manuscript by Tsujino et al. deals with the characterization of emittance and coherence of electron beams emitted from double-gated field emission arrays (FEA). One of the key points is an observed reduction in emittance, achieved by applying a collimation potential to an on-chip gate electrode. Two independent measurements, one of emittance in a DC gun setup and one of beam coherence in a LEED setup, show consistent results. A comparison with theory further confirms the experimental observations.

The presented results are convincing and seem scientifically sound. They are new and original in the sense that they provide evidence that high current and low emittance beams are achievable with double-gated FEAs. However, even though the results do not technically lack novelty, the reported achievements appear to be only a small incremental step compared to earlier publications (refs. 27, 30, 31). The general concept applied here for decreasing emittance has in some related way already been implemented before, and an indication of its basic functionality, namely a reduction in beam divergence has also been found before. Therefore I am hesitant to recommend publication in Nature Communications, and suggest that the authors submit to a more specialized journal.

In case the paper will nevertheless be accepted, there are a few suggestions to improve the manuscript:

- 1) In line 115 V_{ex} is not defined (possibly meaning V_{ge}).
- 2) Line 144 ends with a comma
- 3) In lines 212 & 213 G , d_F , τ_F and b have not been properly described or defined in the text, which makes the equation hard to understand.
- 4) The discussion about laser excitation of the FEAs (line 282) could be enhanced by an estimation of laser parameters required for optical triggering of the gun and achieving similar peak currents (if possible).
- 5) Figure 1 would benefit from a schematic drawing of the structure.
- 6) Adding a comment on whether the solenoid lens in the DC gun test setup could have a detrimental effect on the emittance measurements due to aberrations.

TITLE: Measurement of transverse emittance and coherence of double-gate FEA cathodes
AUTH: S Tsujino, P D Kanungo¹, M Monshipouri, C Lee, R J Dwayne Miller
JOUR: Nature Communications

The manuscript reports on an exciting and very desirable investigation on the emittance properties of double-gated field emitter arrays for high brightness beams used for advanced light sources, an activity for which the authors are very well prepared to undertake. It would be a profound development if field emitters could compete with photocathodes in such applications, and the authors have made good progress in realizing that ambition, with this manuscript building substantially on their earlier work for a fundamentally important metric (emittance). The manuscript appeals to the beam physics, accelerator, and nano-electronics communities, but other communities (such as the e-beam lithography and electron sources communities) will show interest. The authors have been very systematic and clear in their exposition, in highlighting how the advancements will affect various high profile applications, and in relating their theoretical analysis to their data: the very good agreement between straightforward theory and experimental findings was well-done and a pleasure to observe. The references are clear and representative of the literature. The writing style is organized and well-structured. Taken as a whole, the manuscript is a very good work on an important subject that should enjoy interest outside the aforementioned specialized communities. The manuscript is, insofar as this referee can discern, free from error (although a few possible misaligned perceptions could be fixed, as detailed below), and the findings are novel and useful for both application as well as theory and simulation (e.g., likely to be particularly useful to the development of beam optics codes used for accelerator design). The manuscript is therefore recommended for publication.

A few comments and suggestions are made for the benefit of the authors. Text from the manuscript is rendered *blue*:

1. Lines 50-52: when the authors say

yet has not deterred the application of single-tip field emitters as evidenced by their successful use in the high resolution electron microscopy

then they seem to imply that high brightness is achieved in contradiction to a large transverse velocities originating from emission from a highly curved surface, but as the authors' discussion later shows (and their Eq. (3)), the high brightness for a single emitter is *because of the small emission area ($A \propto R^2$) associated with a single emitter tip*, which more than compensates for the large mean transverse energy ($\varepsilon \propto R\sqrt{MTE}$) that otherwise enlarges emittance. There is therefore no contradiction between a single field emitter and high brightness. By comparison, the large emittance for an *array* is due to the footprint of the array. The authors are requested to clarify that fact.

2. Lines 70-71: when the authors say

beam intensity on the beam monitor screen

they seem to be equating emittance with the area of a beam focused onto a screen, which (as they undoubtedly know) is not how emittance is usually understood: a low emittance beam would propagate over a distance without significant spread; a high emittance beam might be

focused to the same area, but would diverge quickly thereafter. That the authors know this is evidenced by their monitoring the rms-beam size over a length. The authors are requested to clarify or emphasize near lines 70-71 that emittance is distinct from focusing to an image plane, that minimal growth *over a propagation distance* is needed.

3. Equation (1) is a minor variation on a commonly used form to infer emission area and field enhancement for field emitters, but there are reasons why this is a flawed procedure [1, 2]. The authors are using it primarily to infer the relationship between field at the tip F_{tip} and gate voltage V_g , but *seem* to do so with a triangular barrier Fowler Nordheim equation (their References [41] and [42]) for current density. This normally would imply that they are neglecting Schottky barrier lowering due to applied field (their value of $b = 6.83 \text{ eV}^{-3/2} \text{ V nm}^{-1}$ confirms this), which I doubt they intended. The authors' approach, however, does follow from Eqs. (15) and (37) of their Ref. [14], as a consequence of approximating the $v(y)$ function by the so-called "Forbes approximation" cited therein. Therefore, the authors could argue that they are *implicitly* using such an approach, but the present discussion creates an erroneous impression when they cite Fowler and Nordheim's 1928 paper. It is recommended that the authors clarify that $b\phi^{3/2}/F_{tip}$ is usable because the Forbes approximation is made in conjunction with a total current tip model, not because an analogy is being drawn between $I(V)$ and $J(F)$ from 1928 FN equation. It would not hurt to mention that $\ln I(V)/V^p$ is linear for p near, but not necessarily equal to, 2.
4. Line 129: *between beamlet emitted* should be *between beamlets emitted*
5. A suggestion regarding Eqs. (7) and (8): The authors are likely well aware that $T(E) \sim P(E)e^{-G(E)}$ where $P(E)$ is a weakly varying function of energy (even Fowler and Nordheim's 1928 paper makes this point). The authors might be on better footing to use $1/d_F = -\partial_E \ln(T(E))$ and then observe that $-\partial_E \ln(P(E))$ is negligible (although not zero), instead of Eq. (8). The results will be the same.
6. Line 302 *The emitter substrate, Em, and* - should E_m be E_m ?
7. Ref. [1] appears to contain a spurious character in the shape of a square in *thin film*.

References

- [1] D. Nicolaescu. "Physical Basis for Applying the Fowler-Nordheim J-E Relationship to Experimental IV Data." J. Vac. Sci. Technol. B, **11(2)**, 392, 1993.
- [2] R.G. Forbes, J.H.B. Deane, N. Hamid, and H.S. Sim. "Extraction of Emission Area From Fowler-nordheim Plots." J. Vac. Sci. Technol. B, **22(3)**, 1222, 2004.

Response to remarks by Reviewer 2

1) In line 115 V_{ex} is not defined (possibly meaning V_{ge}).

2) Line 144 ends with a comma

The typos are corrected.

3) In lines 212 & 213 G , d_F , τ_F and b have not been properly described or defined in the text, which makes the equation hard to understand.

We rewrote the description of the equations and clarified the quantities.

4) The discussion about laser excitation of the FEAs (line 282) could be enhanced by an estimation of laser parameters required for optical triggering of the gun and achieving similar peak currents (if possible).

The laser excitation of FEAs is an important subject. Its discussion requires detailed analysis of the laser excitation condition including the nanoscale electromagnetic distribution and tip-gate interaction (such as the surface-plasmon-enhanced near infrared excitation of the metal nanotips), the estimate of the transient electron distribution that might be considered as non-Fermi-Dirac when the duration of the excitation pulse is very short (ps or below) as usually considered, and the possible modification of the electron energy and moment relaxation path at the metal nanotip geometry. As such, we consider this is out of the scope of the present manuscript.

5) Figure 1 would benefit from a schematic drawing of the structure.

We added a schematic drawing of the structure as Figure 1(c).

6) Adding a comment on whether the solenoid lens in the DC gun test setup could have a detrimental effect on the emittance measurements due to aberrations.

We analyzed the problem and estimated that the nonlinearity of the solenoid lens of the DC gun test setup does not have detrimental effect on the emittance measurements. We added a comment on it in the main text and added a paragraph on the numerical detail in the Supplementary File. In short, we found that we can neglect the saturation of the iron core (because of the low excitation current) and the chromatic aberration (because of the small energy spread of ~ 0.1 eV for the beam energy of 20 keV). As for the spherical aberration, we now treated it explicitly following the reference in the SI and found that the spherical aberration is not important either in our measurement.

Response to remarks by Reviewer 3

1. Clarification of the emittance of the single-tip emitter (around lines 50-52)

We added a comment on the smallness of the emission area of single-tip emitters as the main reason of their small emittance.

2. Clarification of the limitation of our previous measurement in terms of evaluation double-gate FEA's intrinsic emittance (Lines 70-71)

We rewrote the mentioned part to state the limitation of our previous measurement explicitly and in a technically correct way. We believe that the need of the emittance measurement of the present manuscript is better clarified as mentioned by the reviewer.

3. On the description and use of the "Fowler-Nordheim" equation (Equation (1))

We rewrote (1) to a general form and added the explanation as suggested by Reviewer 3; We highlighted the recent important finding by Forbes and Deane in Ref. [42] on the power of the electric field/voltage of the prefactor of the field emission current.

4. Line 129: between beamlet emitted should be between beamlets emitted

The typo is corrected.

5. A suggestion regarding Eqs. (7) and (8):

Following the reviewer's suggestion, we modified equations (7) and (8) and the associated description of the theory to include the slowly varying pre-factor of the transmission function. This way, the discussion became more general mathematically. This has no numerical impact on the previous conclusions as pointed out also by the reviewer.

6. Line 302 The emitter substrate, E_m , and - should E_m be E_m ?

We denote the emitter substrate as E_m , without subscripting "m". No modification was done.

7. Ref. [1] appears to contain a spurious character in the shape of a square in thin lm .

The typo is corrected.

Reviewers' comments:

Reviewer #2 (Remarks to the Author):

The technical issues that were mentioned are now solved.

We maintain, however, the assessment of the initial report (to which the authors have not responded) that this work is incremental and contains no particularly novel physics or breakthrough of technology. From my feeling about what the scope and requirements of Nature Communications are, I cannot recommend publication here.

Reviewer #3 (Remarks to the Author):

The authors have addressed the concerns of this reviewer, and the responses offered to the other reviewer appear reasonable as well and have been carried out with proper diligence. The manuscript continues to be a very good manuscript (now better) and represents an exciting application of field emitters to Free Electron Lasers as well as microscopy. I therefore strongly encourage its publication, as it is timely, consequential and (as a personal observation) exciting.

At this late stage, I do not wish to quibble about minor details that do not affect the findings of the manuscript, and my final comment is simply to clarify an earlier suggestion in the prior review concerning the argument of the exponent in the Fowler Nordheim form of the current density. The authors have (correctly) noted that the power of the electric field in the coefficient of the F^p with $p = 2$ in the FN equation is altered when the Forbes Deane approximation is used (that was the clarification asked for). Therefore, they have implicitly met the request. However, the brevity with which they did so causes me to suspect that they may have overlooked why $b \cdot \phi^{3/2} / F_{tip}$ with $b = 6.83$ works: it is because $v(y) = 1 - y^2 + (1/3) y^2 \ln(y)$, and the \ln term affects the value of p , and leaves the factor of 1 to give the argument the authors give: in other words, it off-loads the Schottky factor to the power of F in the coefficient of J . For the author's benefit, this is discussed in Appendix A.2 of Jensen, K. L., Shiffler, D. A., Petillo, J. J., Pan, Z., Luginsland, J. W. (2014). Emittance, surface structure, and electron emission. *Physical Review Special Topics Accelerators and Beams*, 17(4), 043402-043419. This comment does not entail a requirement on the part of the authors to make changes to the manuscript, as the detail is a fine point and does not affect either the authors conclusions or their treatment, which remain valid.

Reply to the comment by reviewer 2

We thank the reviewer 2 for checking again the technical correctness of our manuscript.

Because of some confusion, we failed to reply to the reviewer 2's remark that I reproduce in the following. We apologize for our misunderstanding and would like to respond to it below.

Reviewer 2 considers, as described in the previous review remark, *the reported achievements appear to be only a small incremental step compared to earlier publications (refs. 27, 30, 31). The general concept applied here for decreasing emittance has in some related way already been implemented before, and an indication of its basic functionality, namely a reduction in beam divergence has also been found before*, and the reviewer 2 maintains it in the present review.

We agree to the reviewer that the reduction in beam divergence itself was already shown by us as well as by other groups, but the qualitative (re-)demonstration of this effect is not what we wish to report in this manuscript.

Our main achievements here are instead,

- The first quantitative measurement of the intrinsic transverse emittance of the double-gate field emitter cathode,
- The first demonstration that the reduced intrinsic transverse emittance value of the double-gate field emitter cathode is in the range competitive to the state-of-the-art photocathode (being used in the accelerators, in the time-resolve electron diffraction experiments, etc.),
- The first theoretical calculation of the intrinsic average transverse beam energy of the field emission beam, not due to the geometrical spread of the electric field and the emission current at the curved emitter tip apexes (as depicted in Ref. 43 in the revised manuscript) but originating from the electron distribution (in free-electron model, that is assumed in the standard field emission theory of metal surfaces), that is compared with experiment.

As described in line 63-67, in our previous imaging experiment in a diode configuration, not only the small beam energy (1-4 keV) made the electron beam susceptible to parasitic fields (magnetic and electric), but also the previous imaging experiment made it impossible to separate the acceleration and the propagation/evolution of the beam, that latter being crucial to quantitatively evaluate the emittance. By implementing a DC gun experiment in which we were able to use the higher (20-40 keV) beam energy and to track the free propagation of the beam, we were able to reveal that a small intrinsic emittance is indeed achieved with our double-gate FEA.

The concept of the double gate field emitter is not new, but its experimental demonstration is achieved by the present work for the first time. With this demonstration, one can further explore the optimization of the field emitters for various applications based on the further engineering of the gate structures or by pursuing new materials guided by theory. We note that the direct validation of the intrinsic emittance by experiment is critically important for applications such as the accelerator applications since construction of expensive apparatus demands precise pre-definition of the cathode characteristics such as the intrinsic emittance, not only estimated theoretically but also measured by experiment.

Further in the present work, we demonstrated,

- The first successful application of the micro- and nano-fabricated field emitter to the low-energy electron diffraction and the successful evaluation of the transverse coherence length,
- The demonstration of the quantitative agreement of the intrinsic emittance and the transverse coherence length.

In summary, we consider these points listed above are significant and we disagree to the reviewer's assessment. Since these were already described in the manuscript, we made no revision of the text.

Reply to the comment by reviewer 3

We thank the reviewer 3 for the additional comment on the exactness of our I-V fitting procedure that lead to the estimation of F_{tip} . I agree to the reviewer that the work by Forbes and Deane lies in the discovery of the ln-term that doesn't affect the exponential term of the "Fowler-Nordheim" current density.

To clarify this point, we added the following comment (red) in the sentence at the line 103 (page 4) together with the reference on the mentioned literature (Ref. 43, Jensen et al., Phys. Rev. STAB), which is read in the revised text as,

The exponential term in (1) **is unaffected by the precise treatment of the image charge effect [42,43] and** gives the dominant contribution to the $I-V_{ge}$ relation.

In the line 210, we also added the citation to Ref. 43 since it exactly discussed it. We also corrected the citation to Ref. 42 (that was left referring to the original Fowler Nordheim article).

REVIEWERS' COMMENTS:

Reviewer #2 (Remarks to the Author):

The authors have explained in their letter in detail why they think their work is novel and substantial. Some of their arguments are valid indeed. While overall I still have some doubts, I could - given the made clarifications - now live well with publication.